# Effects of Low-Molecular-Weight Glutenin Subunit Encoded by *Glu-A3* on Gluten and Chinese Fresh Noodle Quality

**DOI:** 10.3390/foods12163124

**Published:** 2023-08-20

**Authors:** Hongwei Zhou, Yingquan Zhang, Yanning Yang, Yuyan Zhang, Jinfu Ban, Bo Zhao, Lei Zhang, Xiaoke Zhang, Boli Guo

**Affiliations:** 1Institute of Food Science and Technology, Chinese Academy of Agricultural Sciences/Comprehensive Utilization Laboratory of Cereal and Oil Processing, Ministry of Agriculture and Rural, Beijing 100193, China; zhouhw12356@163.com (H.Z.); zhangyingquan@caas.cn (Y.Z.); zhangyy202205@163.com (Y.Z.); zhaobo794@126.com (B.Z.); 15901490010@163.com (L.Z.); 2College of Agronomy, Northwest A & F University, Yangling 712100, China; yangyn17@163.com; 3Shijiazhuang Academy of Agricultural and Forestry Sciences, Shijiazhuang 050041, China; sjznkybjf@163.com

**Keywords:** wheat, LMW-GS, *Glu-A3*, gluten quality, Chinese fresh noodle

## Abstract

Low-molecular-weight glutenin subunits (LMW-GS) account for 40% of the total wheat grain gluten protein fraction, which plays a significant role in the formation of noodle processing quality. The goal of this study was to clarify the effects of the major LMW-GS encoded by *Glu-A3* on gluten and Chinese fresh noodle (CFN) quality. Four near-isogenic lines (NILs) were used as materials in this study, respectively carrying alleles *Glu-A3a*, *Glu-A3b*, *Glu-A3c*, and *Glu-A3e*, against the background of wheat variety Xiaoyan 22. The grain protein and its component contents and the gluten content, gluten index, farinograph properties, cooking quality, and textural quality of CFN were investigated. The results show that the ratios of glutenin to gliadin (Glu/Gli) in the NILs ranked them as *Glu-A3b* > *Glu-A3c*/*Glu-A3a* > *Glu-A3e*, and the unextractable polymeric protein content (UPP%), gluten index (GI), and farinograph quality in the NILs ranked them as *Glu-A3b* > *Glu-A3c* > *Glu-A3a*/*Glu-A3e*. Compared to *Glu-A3b* and *Glu-A3a*, the NILs carrying alleles *Glu-A3c* and *Glu-A3e* had better cooking and texture properties in CFN. All these findings suggest that the introduction of alleles *Glu-A3c* or *Glu-A3e* is an efficient method for quality improvement in CFN, which provides an excellent subunit selection for improving CFN quality.

## 1. Introduction

Wheat (*Triticum aestivum* L.) is one of the three most widely planted and highest-yielding cereal crops in the world and a critical source of energy and nutrients in the human diet. It can be processed into various popular wheat-based food products (bread, cakes, noodles, Chinese steamed bread, biscuits, etc.) due to its gluten protein creating dough with unique viscoelastic properties [1]. Gluten protein accounts for about 80% of the total protein of wheat. Gluten protein is composed of high-molecular-weight glutenin subunits (HMW-GS), low-molecular-weight glutenin subunits (LMW-GS), and gliadin [2]. The LMW-GS account for 40% of the total grain gluten protein fraction, which plays a significant role in the end-use quality [3,4]. However, research on the relationship between the LMW-GS and the end-use quality is limited mainly because of the complexity at both the gene and protein levels [5].

Wheat LMW-GS are encoded by three loci (*Glu-A3*, *Glu-B3*, *Glu-D3*) on the short arms of group-1 homologous chromosomes (1A, 1B, and 1D) (Payne 1987). Previously, the nomenclature of *Glu-3*-encoding LMW-GS was not consistent among laboratories due to the complexity of the LMW-GS and the distinct separation methods used by different researchers, which hampered the sharing of information about the effects of individual LMW-GS on quality properties [6]. In 2008, an international unified nomenclature system for *Glu-3* alleles was established, naming eight alleles at the *Glu-A3* locus, twenty alleles at the *Glu-B3* locus, and nine alleles at the *Glu-D3* locus in common wheat [6]. The effects of different *Glu-3* loci on flour-processing qualities have been ranked in bread wheat. *Glu-A3* and *Glu-B3* alleles are supposedly more important than *Glu-D3* alleles in terms of their end-use processing qualities [7]. At the *Glu-A3* locus, the deletion of LMW-GS *Glu-A3a* significantly reduced dough strength and breadmaking quality [8]. *Glu-A3e* was a favorable allele for dough-mixing properties [9], whereas a contradictory result was reported showing that *Glu-A3e* performed the worst in dough properties and bread quality [10]. At the *Glu-B3* locus, the deletion of LMW-GS *Glu-B3h* significantly reduced dough strength and breadmaking quality [7]. *Glu-B3b*, *Glu-B3g*, and *Glu-B3i* were more highly correlated with superior breadmaking quality than other *Glu-B3* alleles [9], but *Glu-B3b* and *Glu-B3i* being related to better dough properties has not been reconfirmed [11].

The importance of LMW-GS on the quality of flour-based food products has been widely reported. According to the different processing methods, flour-based food products can be divided into steamed/boiled foods (Chinese steamed bread, noodles, etc.) and baked foods (bread, cookies, cakes, etc.). However, recent research on LMW-GS has mainly focused on its effect on breadmaking quality [7,8,10]. Noodles are the main traditional staple food, accounting for approximately 40% of the total wheat consumed in Asia [12]. As living standards rapidly increase, the genetic improvement of wheat for noodle qualities has become increasingly important. He et al. [13] reported that the alleles *Glu-A3d* and *Glu-B3d* were slightly better than others for dry white Chinese noodle quality. Jin et al. [11] reported that the line with a glutenin composition (1, 7 + 9, 2 + 12, *Glu-A3c*, *Glu-B3d,* and *Glu-D3c*) had superior viscoelasticity in raw white Chinese noodles.

Although it is well known that LMW-GS affect flour-based food-processing quality, according to previous studies, the effects of some *Glu-3* alleles on dough and bread properties are contradictory. Thus, much more effort is also needed to understand the associations among LMW-GS, dough properties, and end-use quality, especially with the same genetic background. This is now possible, thanks to the availability of validated molecular markers for alleles encoded by *Glu-A3* [14,15]. Meanwhile, the applicability of LMW-GS to traditional Chinese steamed food deserves further study. Near-isogenic lines (NILs) are groups of materials with the same genes and similar genetic backgrounds except for the target genes, and they are ideal materials for precisely exploring quality differences between LMW-GS allelic variants [16]. In this study, a set of NILs respectively containing LMW-GS alleles *Glu-A3a*, *Glu-A3b*, *Glu-A3c*, and *Glu-A3e* was used to study the effects of LMW-GS on gluten and Chinese fresh noodle (CFN) properties. The results will provide a theoretical reference for wheat breeding and the processing of high-quality CFN.

## 2. Materials and Methods

### 2.1. Plant Materials

Four BC_6_F_3_ NILs were used as materials in this study, respectively carrying LMW-GS alleles *Glu-A3a*, *Glu-A3b*, *Glu-A3c*, and *Glu-A3e* against the background of Xiaoyan 22. They were obtained by one hybridization, six backcrosses, and three subsequent self-crosses. Sequence-tagged site (STS) molecular markers were applied for selecting the target LMW-GS in the cross process.

All of the NILs were planted in field plots at Yangling (108°40′ E, 34°160′ N), Shaanxi Province of China, in 2020/2021. Two biological replications were used for each material, and each replication had 10.4 m^2^. Three-element complex fertilizer containing 22% nitrogen (N), 14% phosphatic, and 6% kalium with 750 kg/ha and commercial granular organic fertilizer containing 45% organic matter with 900 kg/ha were applied before plowing.

After harvest, the grains were dried and stored for at least 60 days before milling and testing. The grains were tempered for 24 h at a 15.5% (*w*/*w*) moisture level and milled in an MLU 202 Buhler experimental mill (Buhler, Uzwil, Switzerland).

### 2.2. Identification of Gluten Composition

#### 2.2.1. Identification of LMW-GS Composition

Firstly, the genomic DNA of four lines was extracted from the wheat leaves using the cetyltrimethylammonium bromide (CTAB) method. Subsequently, the target sequence was amplified by polymerase chain reaction (PCR) in the genome according to the developed markers. Then, the amplified products were separated by agarose gel. Finally, the amplified products after separation were identified according to the fragment size (Table 1).

#### 2.2.2. Identification of HMW-GS Composition

The HMW-GS of the four lines was extracted and fractionated according to the method described by Wang et al. [17]. Specifically, the grain glutenin was extracted using a solution containing 0.0625 mol/L Tris-HCL (pH 6.8), 2% SDS, 5% 2-Hydroxy-1-ethanethiol, 10% glycerol, and 0.05% bromophenol blue. Then, the glutenin was separated by sodium dodecyl sulfate polyacrylamide gel electrophoresis (SDS-PAGE).

#### 2.2.3. Identification of Gliadin Composition

The gliadin of the four lines was extracted and fractionated according to the method described by Wang et al. [17]. Specifically, the grain gliadin was extracted using 70% (*v*/*v*) aqueous ethanol. Then, the gliadin was separated by acid polyacrylamide gel electrophoresis (A-PAGE).

### 2.3. Determination of Grain Protein Content

The grain protein content (GPC) was evaluated according to the AACC method 46-30 (2000). Specifically, grain samples were ground into flour using a MM 400 mixer mill (Retsch, Haan, German). Then, the GPC was detected by a DN2100 Dumas (Beijing Nordtec Instrument Co., Ltd., Beijing, China).

### 2.4. Grain Protein Extraction, Separation, and Quantification

First, 1 mL of phosphate buffer solution (PBS) (0.05 M, pH 7.0, 0.5% SDS) was added to about 20 mg of whole wheat flour without sonication. Then, the supernatant was collected as SDS-extractable proteins. Another 1 mL of PBS extraction buffer was added to the precipitate for the extraction of the SDS-unextractable proteins. The pellet was suspended in the solution and sonicated for 2 min (30 rounds of 2-s pulses and 2-s intervals) at 30 W of power. Afterward, the supernatant was collected as SDS-unextractable proteins. All extracts (supernatant) were filtered through a 0.45 µm polyvinylidene fluoride (PVDF). The separation and quantification of the SDS-extractable proteins and SDS-unextractable proteins were performed by SE-HPLC using an Agilent 1260 LC system (Agilent, Palo Alto, USA). Twenty microliters of the extractions were injected into a TSKgel G4000SWXL (7.8 × 300 mm) column (Tosoh Corporation, Tokyo, Japan) maintained at 30 °C. The flow rate was adjusted to 0.5 mL/min. The proteins were separated by using a constant gradient with PBS over 30 min and detected by UV absorbance at 214 nm.

The proteins were roughly divided into three fractions, i.e., glutenin (Glu), gliadin (Gli), and albumin and globulin (Alb/Glo), respectively. The SDS-extractable and SDS-unextractable Glu, Gli, and Alb/Glo were designated as EGlu and UGlu, EGli and UGli, and EAlb/Glo and UAlb/Glo, respectively. The contents of the different protein fractions were calculated as follows:Total area = EGlu + UGlu + EGli + UGli + EAlb/Glo + UAlb/Glo
Glutenin content = (EGlu + UGlu)/Total area × 100
Gliadin content = (EGli + UGli)/Total area × 100
Albumin and globulin content = (EAlb/Glo + UAlb/Glo)/Total area × 100
Glutenin/gliadin = (EGlu + UGlu)/(EGli + UGli)
Unextractable polymeric protein (UPP) content = UGlu/(EGlu + UGlu) × 100

### 2.5. Determination of Gluten Content and Index

The wet gluten content (WGC), dry gluten content (DGC), and gluten index (GI) of the flour were measured using handwashing according to Chinese standard GB/T 5506.1-2008, and corrected to a 14% moisture content.

### 2.6. Farinograph Analysis

The dough rheological parameters were evaluated according to the AACC method 54-21 (2000) using a farinograph (Brabender, Duisburg, Germany) equipped with a 300 g bowl. The water absorption (WA), dough development time (DDT), stability time (ST), degree of softening (DS), and farinograph quality number (FQN) were determined.

### 2.7. Determination of Chinese Fresh Noodle (CFN) Quality

According to the method by Zhang et al. [18] with some modifications, CFN was prepared, and the optimal cooking time (OCT), water absorption rate (WAR), cooking loss ratio (CLR), and texture characteristics of the cooked CFN were determined. 

#### 2.7.1. Preparation of CFN

For the preparation of the CFN, the ingredients (200 g flour, 35% water, 1% salt) were mixed for 4 min to make dough crumbs, and the obtained dough crumbs were rolled twice on a JMTD-168/140 sheet rolling machine (Beijing Dongfu Jiuheng Instrument Technology Co., LTD., Beijing, China) to obtain a 2.5 mm thick dough sheet, which was put into zip-lock bags, sealed, and rested at 25 °C for 30 min. Thereafter, the dough sheet was continuously rolled four times to obtain a 1 mm thick sheet, which was then cut into 1 mm wide wet raw noodles.

#### 2.7.2. Determination of Cooking Characteristics of CFN

For the determination of the OCT, 10 CFN sticks were placed into 500 mL of boiling distilled water. When the white core disappeared, they were removed from the heat, and the time was recorded as the OCT of the CFN.

For the determination of the WAR and CLR, 10.00 g of noodles was weighed, and 500 mL of distilled water was added to a stainless steel pot and heated to the boiling point with an induction cooker. The weighed sample was added to the pot, and the boiling state was maintained. After the OCT was reached, the noodles were obtained from the pot immediately. When no obvious water could be observed on the noodle surface, the noodle was weighed to calculate the WAR. The remaining noodle soup in the stainless-steel pot was heated until all liquid had evaporated. The stainless steel pot was put into an oven and baked at 130 °C to a constant weight. The total mass of the stainless steel pot and the remaining material was weighed to calculate the CLR.

#### 2.7.3. Determination of Texture Characteristics of CFN

For the determination of the texture characteristics, twenty noodles were randomly selected, cooked for the OCT, fished out immediately, and rinsed with tap water for 20 s. A set of five sticks were placed in parallel on the loading platform in a TA.XT plus Texture Analyzer (Stable Micro System, Godalming, UK). The testing parameters were set as follows: TPA measurement mode; probe, A/LKB-F; speed before the measurement, 2 mm/s; measurement speed, 0.8 mm/s; speed after the measurement, 2 mm/s; compression ratio, 70%; interval between two compressions, 10 s; starting induction force, initial value, 10 g; and data acquisition rate, 200 pps. Three parallel experiments were performed for each sample.

### 2.8. Statistical Analysis

All of the analyses in this study were performed at least in triplicate. One-way analysis of variance (ANOVA) followed by Duncan’s test was conducted to verify significant differences (*p* < 0.05) using SPSS 22.0 software for Windows (SPSS Inc., Chicago, IL, USA).

## 3. Results and Discussion

### 3.1. Identification of Gluten Composition

#### 3.1.1. LMW-GS Composition

The developed markers were used to identify the LMW-GS composition encoded by the *Glu-A3* and *Glu-B3* loci of four wheat lines (Table 1). The fragments of 529 bp (lane 2), 894 bp (lane 3), 573 bp (lane 4), and 158 bp (lane 6) were obtained from the four wheat lines using markers Glu-A3a, Glu-A3b, Glu-A3ac, and Glu-A3e, respectively (Figure 1A), which indicates that the four NILs respectively contained LMW-GS alleles *Glu-A3a*, *Glu-A3b*, *Glu-A3a*/*Glu-A3c*, and *Glu-A3e*. To further determine the LMW-GS composition of NIL *Glu-A3c*, no fragment (lane 5) was determined using marker Glu-A3a (Figure 1A), which indicates that the NIL *Glu-A3c* only contained allele *Glu-A3c*. In addition, by using marker Glu-B3j, the four wheat lines all obtained the fragment of 1500 bp (Figure 1B, lanes 2, 3, 4, and 5), which indicates that the four lines had identical LMW-GS at the *Glu-B3* locus.

#### 3.1.2. HMW-GS and Gliadin Composition

SDS-PAGE and A-PAGE were respectively used to confirm the HMW-GS and gliadin composition of the four wheat lines. Subunit N at the *Glu-A1* locus, 7 + 9 at the *Glu-B1* locus, and 2 + 12 at the *Glu-D1* locus were confirmed in all four wheat lines (Figure 2A). Meanwhile, identical gliadin compositions were also confirmed in all four wheat lines (Figure 2B).

These results show that the four wheat lines had identical gluten compositions, except for the target LMW-GS (*Glu-A3a*, *Glu-A3b*, *Glu-A3c*, and *Glu-A3e*).

### 3.2. Effects of LMW-GS on Grain Protein and Its Component Contents

The grain protein content (GPC) and its component contents can influence the dough properties and the quality of flour-based food products [19,20]. As shown in Figure 3A, the GPC was not significant different among the NIL *Glu-A3a*, *Glu-A3b*, *Glu-A3c* and *Glu-A3e*, with average values of 15.31%, 15.24%, 15.23%, and 14.84%, respectively. To analyze the grain protein component content, SE-HPLC was used to determine the SDS-extractable and SDS-unextractable glutenin, gliadin, and albumin/globulin contents. The glutenin contents of the four NILs ranked them as *Glu-A3b*(40.24%) > *Glu-A3a*(38.72%)/*Glu-A3c*(38.96%) > *Glu-A3e*(36.36%). On the contrary, the gliadin contents of the four NILs ranked them as *Glu-A3e*(59.89%) > *Glu-A3a*(57.83%)/*Glu-A3c*(57.04%) > *Glu-A3b*(55.65%). Meanwhile, the albumin and globulin contents had no significant differences among the four NILs (*p* > 0.05, Figure 3B). Complementation between the glutenin content and gliadin content ensured the stability of the total grain protein content among the four NILs. These alterations in the glutenin and gliadin contents caused significant differences in the ratios of glutenin to gliadin (Glu/Gli). These significant differences in the Glu/Gli of the NILs ranked them as *Glu-A3b*(0.72) > *Glu-A3c*(0.68)/*Glu-A3a*(0.67) > *Glu-A3e*(0.61) (*p* < 0.05, Figure 3C). This indicates that LMW-GS changed the gluten component content. In addition, the unextractable polymeric protein content (UPP%) of the NILs ranked them as *Glu-A3b*(35.00%) > *Glu-A3c*(32.03%) > *Glu-A3a*(29.55%)/*Glu-A3e*(29.04%) (*p* < 0.05, Figure 3D). A lower UPP% occurred in NIL *Glu-A3e*, which agrees with the study of Zhang et al. [10]. Generally, higher Glu/Gli promotes the formation of more UPP [21]. Compared to *Glu-A3e*, the NILs carrying allele *Glu-A3b* or *Glu-A3c* had a higher UPP% mainly because of the higher Glu/Gli, which indicates that a superior LMW-GS encoded by *Glu-A3* promotes more UPP formation by improving the Glu/Gli. However, the Glu/Gli and UPP% between the NILs *Glu-A3a* and *Glu-A3e* did not agree with this rule. Li et al. [22] showed that HMW-GS/LGW-GS also effect the formation of UPP. A potential difference in the HMW-GS/LMW-GS between *Glu-A3a* and *Glu-A3e* may be the reason for the same UPP%.

### 3.3. Effects of LMW-GS on Gluten Content and Index

As variations in the protein component content normally affect the gluten quality [23], we further measured the gluten content and index (Figure 4). The wet gluten content (WGC) and dry gluten content (DGC) in the flours ranged from 34.71% to 35.85% and from 11.61% to 11.96%, respectively, and no significant differences were found among the four NILs (*p* > 0.05, Figure 4A,B). Gluten is mainly made up of glutenin and gliadin. The same dry gluten content among the four NILs confirms the result of the grain gluten (glutenin plus gliadin) content. Meanwhile, the gluten index (GI) in the NILs ranked them as *Glu-A3b*(43.74%) > *Glu-A3c*(40.56%) > *Glu-A3a*(37.35%)/*Glu-A3e*(37.35%) (*p* < 0.05, Figure 4C). Thus, the different LMW-GS encoded by *Glu-A3* facilitated the formation of different degrees of complex and compact gluten networks displaying significantly different GIs. 

### 3.4. Effects of LMW-GS on Farinograph Parameters

The water absorption (WA), dough development time (DDT), stability time (ST), and farinograph quality number (FQN) were positively correlated to the dough strength, while the degree of softening (DS) was negatively correlated to the dough strength. Notably, as shown in Table 2, the DDT, ST, DS, and FQN had bigger coefficients of variation and exhibited the same or opposite variation trends among the four NILs. The DDT, ST, and FQN of the four NILs ranked them as *Glu-A3b* > *Glu-A3c* > *Glu-A3a* > *Glu-A3e*. Conversely, the DS of the four NILs ranked them as *Glu-A3e* > *Glu-A3a* > *Glu-A3c* > *Glu-A3b*. These results reconfirm the differences in the gluten networks among the four NILs in a wheat dough system. Previous studies on the influence of allele *Glu-A3e* on dough characteristics have been contradictory [9,10]. Our results show that allele *Glu-A3e* was related to inferior dough characteristics.

According to the results of the gluten index and farinograph properties, obvious differences in gluten strength in the NILs ranked them as *Glu-A3b* > *Glu-A3c* > *Glu-A3a*/*Glu-A3e*. Considering the overall worse gluten quality against the genetic background of Xiaoyan 22, we did not evaluate the quality of the bread of the four NILs. Generally speaking, higher gluten strength contributes to better-quality bread, which indicates that the quality of bread with the different LMW-GS alleles could rank them as *Glu-A3b* > *Glu-A3c* > *Glu-A3a*/*Glu-A3e*.

### 3.5. Effects of LMW-GS on Quality Characteristics of Chinese Fresh Noodles (CFNs)

Noodles are the main traditional staple food, and they play an essential role in people’s daily diet. Excavating superior LMW-GS is an important foundation for the improvement of noodle quality. Meanwhile, the relationship between gluten strength and noodle quality is worth further investigation [24,25]. Thus, we further investigated the effect of LMW-GS on cooking and the texture characteristics of CFNs.

#### 3.5.1. Cooking Characteristics of CFNs

The cooking quality refers to the capability to withstand disintegration upon prolonged cooking, together with a satisfactory degree of cooking and tenderness in the final product. A good-quality noodle should have a short cooking time with little loss of solids and high water absorption in the cooking [26,27]. The optimal cooking time (OCT), water absorption ratio (WAR), and cooking loss ratio (CLR) are presented in Table 3. The OCT was consistent among the four NILs. The WAR of the noodles made with NIL *Glu-A3a* was obviously lower than those made with *Glu-A3b*, *Glu-A3c*, and *Glu-A3e*, while no significant differences in the WAR were found among the noodles respectively made with NIL *Glu-A3b*, *Glu-A3c*, and *Glu-A3e*. The CLR of the noodles made with NIL *Glu-A3c* was significantly higher than that of the *Glu-A3a* noodles, while there was no difference from that of the *Glu-A3e* noodles. Meanwhile, the CLR of the noodles made with NIL *Glu-A3b* was obviously higher than those of the other three NIL noodles. The structure of noodles is mainly composed of protein–starch networks. In this study, the difference in the materials was only in the LMW-GS composition at the *Glu-A3* locus, and this difference further resulted in the differences in Glu/Gli and gluten strength. Generally speaking, superior-quality proteins strongly hold the starch molecules, and therefore, prevent the excessive swelling, rupture, and leaching of starch during cooking, facilitating lower water absorption and cooking loss [28]. However, our results show that the noodles made with NIL *Glu-A3b* had the highest gluten strength, highest WAR, and maximum CLR, which may be related to the exorbitant Glu/Gli or gluten strength. Gao et al. [29] indicated that there is little gliadin to fill and adhere to the gluten network when the Glu/Gli is too high, and the gluten network will be weak and easy for water molecules to invade and for solids to be lost. Hence, a suitable Glu/Gli and gluten strength is critical for the making of high-quality noodles [29,30]. A better cooking property was found in the noodles made with NIL *Glu-A3e* or *Glu-A3c*, showing a low to moderate gluten strength in our study.

#### 3.5.2. Texture Characteristics of CFN

The texture analyzer is a widely used instrument for objectively evaluating noodles’ edible quality [12]. Compared to sensory evaluation, it has good reproducibility in evaluating the hardness, adhesiveness, springiness, resilience, and chewiness of cooked noodles [31]. The texture properties of cooked noodles are among the primary concerns of consumers [32]. The results of the texture properties are presented in Table 4. The hardness, adhesiveness, springiness, cohesiveness, and resilience of cooked noodles carrying different LMW-GS ranked them as *Glu-A3a* > *Glu-A3b* > *Glu-A3c*/*Glu-A3e*, *Glu-A3e*/*Glu-A3c* > *Glu-A3b*/*Glu-A3a*, *Glu-A3b*/*Glu-A3c*/*Glu-A3e* > *Glu-A3a*, *Glu-A3c*/*Glu-A3*e > *Glu-A3b* > *Glu-A3a*, and *Glu-A3c*/*Glu-A3e* > *Glu-A3b* > *Glu-A3a*, while no significant difference in chewiness was found among the four NILs. The lower the water content of the noodles, the higher the hardness, as reported by Sangpring et al. [33], which may explain the highest hardness of the noodles carrying LMW-GS allele *Glu-A3a*, which had the lowest water absorption ratio of cooked CFNs and the same water contents of CFNs with other NILs (not listed). According to the standard of GB/T 35875-2018, a superior cooked noodle should have suitable hardness, lower adhesiveness, higher elasticity, and chewiness. Our results show that the cooked noodles made with NIL *Glu-A3c* or *Glu-A3e* had better texture characteristics, while those of *Glu-A3a* or *Glu-A3b* had inferior adhesiveness and elasticity. However, using the sensory evaluation method, Jin et al. [11] found that the hardness, viscoelasticity, and smoothness of cooked noodles carrying different LMW-GS encoded by *Glu-A3* had no significant difference. This difference in the results may be related to the difference in the materials or estimation method.

A close association between the gluten strength and Chinese dry white noodle quality was observed by Liu et al. [25], but this was not confirmed by a study of gluten strength and Chinese fresh white noodles [24]. Neelam et al. [28] showed that medium strong flours were found to be most suitable for instant noodle preparation. He et al. [34] showed that medium protein content and medium-to-strong gluten strength with good extensibility were desirable for mechanized dough production, but weak-to-medium gluten strength was better for manual production. In this study, the noodles made with NIL *Glu-A3e* or *Glu-A3c* with a low to moderate gluten strength had better cooking and texture properties. Therefore, suitable gluten quality is a very important factor in determining noodle quality.

## 4. Conclusions

In the present study, we dissected the protein composition of four NILs and measured the protein and its component contents and the gluten content, gluten index, farinograph properties, and Chinese fresh noodle (CFN) quality of individual NILs. The four NILs had the same protein composition (except the target LMW-GS), protein content, and gluten content, while the Glu/Gli in the NILs ranked them as *Glu-A3b* > *Glu-A3c*/*Glu-A3a* > *Glu-A3e*, and their UPP%, GI, and farinograph quality ranked them as *Glu-A3b* > *Glu-A3c* > *Glu-A3a*/*Glu-A3e*. The NIL carrying *Glu-A3b* showed higher Glu/Gli, UPP%, GI, and farinograph quality. It is possible that LMW-GS alleles determine gluten and dough quality by improving the Glu/Gli and further modifying the size distribution of glutenin polymers. Meanwhile, the CFNs made by NIL *Glu-A3c* or *Glu-A3e* with low to moderate Glu/Gli and gluten strength had superior cooking and texture properties. These results significantly enhance our understanding of the gluten and Chinese fresh noodle effects of LMW-GS, and provide appropriate LMW-GS selections for wheat breeding and processing of high-quality CFNs.

## Figures and Tables

**Figure 1 foods-12-03124-f001:**
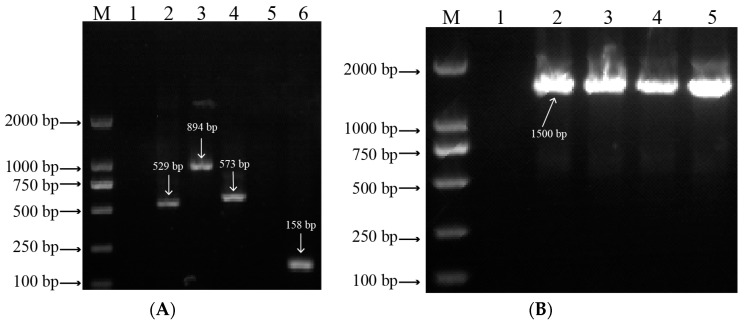
Marker detection of allelic variation of LMW-GS encoded by *Glu-A3* and *Glu-B3* loci in agarose gel electrophoresis. (**A**) M, DNA marker 2000; lane 1, water; lane 2, *Glu-A3a* (529 bp); lane 3, *Glu-A3b* (894 bp); lane 4, *Glu-A3c* (573 bp); lane 5, *Glu-A3c*; lane 6, *Glu-A3e* (158 bp). (**B**) M, DNA marker 2000; lane 1, water; lane 2, *Glu-A3a* (1500 bp); lane 3, *Glu-A3b* (1500 bp); lane 4, *Glu-A3c* (1500 bp); lane 5, *Glu-A3e* (1500 bp).

**Figure 2 foods-12-03124-f002:**
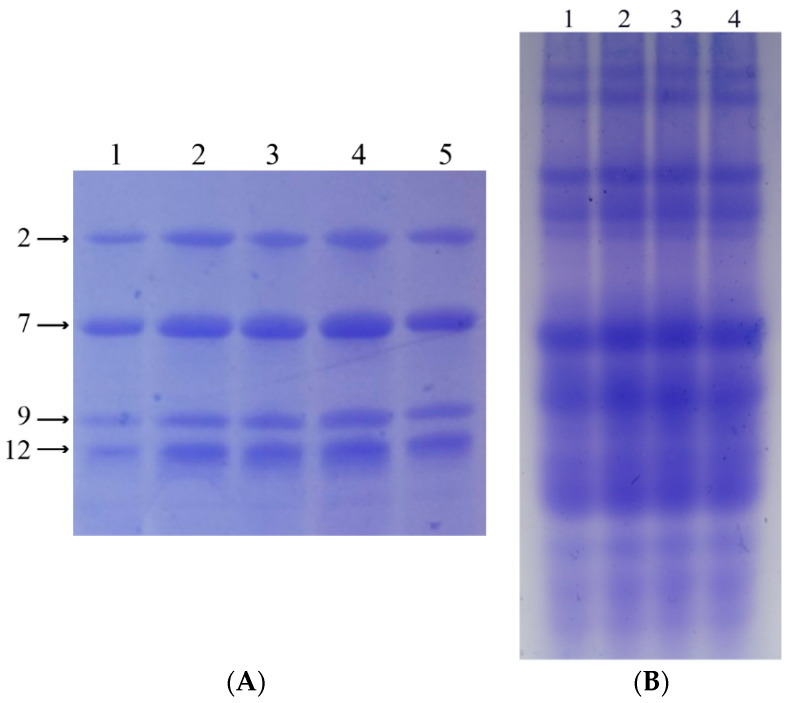
Identification of HMW-GS and gliadin composition of four lines. (**A**) HMW-GS composition of Xioayan 22 and four lines. Lane 1, Xiaoyan22 (null/7 + 9/2 + 12); lane 2, *Glu-A3a*; lane 3, *Glu-A3b*; lane 4, *Glu-A3c*; lane 5, *Glu-A3e*. (**B**) Gliadin composition of Xiaoyan 22 and four lines. Lane 1, *Glu-A3a*; lane 2, *Glu-A3b*; lane 3, *Glu-A3c*; lane 4, *Glu-A3e*.

**Figure 3 foods-12-03124-f003:**
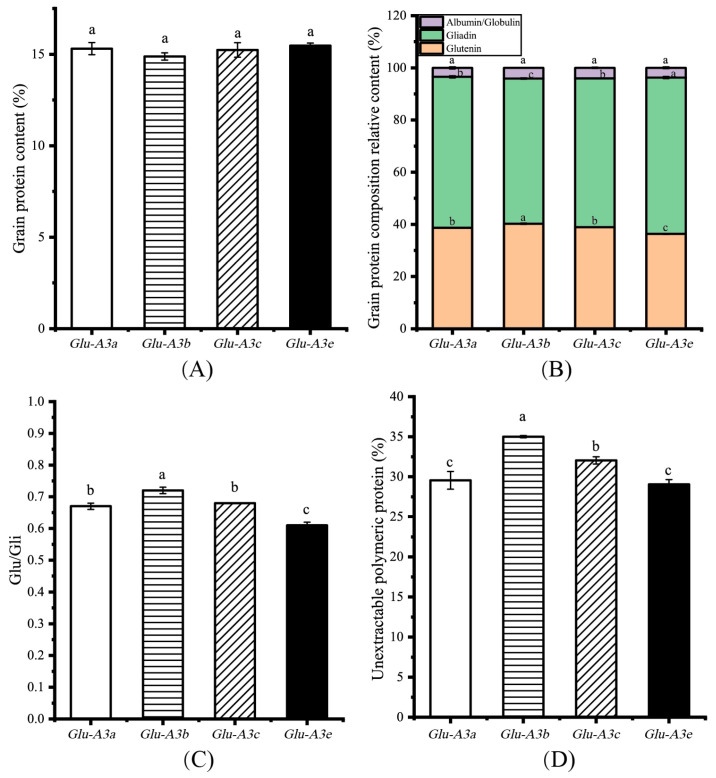
Effects of the four NILs carrying different LMW-GS encoded by *Glu-A3* on the grain protein composition. (**A**) Grain protein content (%); (**B**) grain protein component content (%); (**C**) Glu/Gli; (**D**) unextractable polymeric protein content (%). Different lowercase letters indicate significant differences among the four NILs at *p* < 0.05.

**Figure 4 foods-12-03124-f004:**
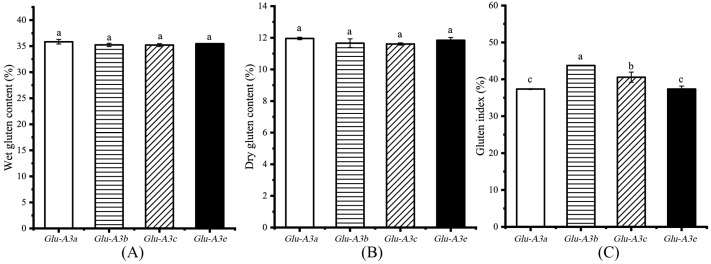
Effects of the four NILs carrying different LMW-GS encoded by *Glu-A3* on the flour gluten content and index. (**A**) Wet gluten content (%); (**B**) dry gluten content (%); (**C**) gluten index (%). Different lowercase letters indicate significant differences among the four NILs at *p* < 0.05.

**Table 1 foods-12-03124-t001:** PCR primers of markers used for the discrimination of alleles *Glu-A3a*, *Glu-A3b*, *Glu-A3c*, *Glu-A3e*, and *Glu-B3j*.

Marker	Sequence (5′→3′)	Fragment Size (bp)	Reference
*Glu-A3a*	F: AAACAGAATTATTAAAGCCGG R: GGTTGTTGTTGTTGCAGCA	529	Jin et al. [14]; Wang et al. [15]
*Glu-A3b*	F: TTCAGATGCAGCCAAACAA R: GCTGTGCTTGGATGATACTCTA	894	
*Glu-A3ac*	F: AAACAGAATTATTAAAGCCGG R: GTGGCTGTTGTGAAAACGA	573	
*Glu-A3e*	F: AAACAGAATTATTAAAGCCGG R: GGCACAGACGAGGAAGGTT	158	
*Glu-B3j*	F: GGAGACATCATGAAACATTTG R: CTGTTGTTGGGCAGAAAG	1500	

**Table 2 foods-12-03124-t002:** Effects of the four NILs carrying different LMW-GS encoded by *Glu-A3* on farinograph parameters.

NIL	*Glu-A3a*	*Glu-A3b*	*Glu-A3c*	*Glu-A3e*	CV (%)
Water absorption (%)	69.2	69.0	69.6	68.8	0.5
Dough development time (min)	2.4	3.1	3.0	2.2	16.5
Stability time (min)	1.0	1.5	1.3	0.9	23.4
Degree of softening (BU)	165	104	137	175	21.9
Farinograph quality number (mm)	31	44	39	29	19.6

**Table 3 foods-12-03124-t003:** Effects of the four NILs carrying different LMW-GS encoded by *Glu-A3* on quality characteristics of CFN.

NIL	Optimal Cooking Time (s)	Water Absorption Ratio (%)	Cooked Loss Ratio (%)
*Glu-A3a*	180	87.98 ± 3.35 b	5.54 ± 0.48 c
*Glu-A3b*	180	93.71 ± 2.48 a	9.49 ± 0.35 a
*Glu-A3c*	180	93.35 ± 2.99 a	6.11 ± 0.38 b
*Glu-A3e*	180	95.26 ± 1.89 a	5.91 ± 0.35 bc

The data followed by different letters in the same column mean significant differences (*p* < 0.05).

**Table 4 foods-12-03124-t004:** Effects of the four NILs carrying different LMW-GS encoded by *Glu-A3* on TPA characteristics of CFN.

NIL	Hardness (g)	Adhesiveness (g×s)	Springiness (%)	Cohesiveness (%)	Resilience (%)	Chewiness (g)
*Glu-A3a*	284.58 ± 10.96 a	−3.01 ± 1.43 b	84.12 ± 1.84 b	62.34 ± 1.81 c	35.81 ± 2.02 c	152.47 ± 4.83 a
*Glu-A3b*	266.87 ± 11.47 b	−3.18 ± 1.27 b	87.08 ± 1.56 a	64.23 ± 1.57 b	37.55 ± 2.46 b	151.72 ± 8.61 a
*Glu-A3c*	236.99 ± 13.47 c	−1.23 ± 0.36 a	87.60 ± 0.64 a	71.93 ± 1.15 a	46.24 ± 1.55 a	148.44 ± 6.90 a
*Glu-A3e*	238.72 ± 6.77 c	−1.14 ± 0.24 a	87.57 ± 0.89 a	72.84 ± 1.10 a	47.16 ± 1.15 a	151.01 ± 3.63 a

The data followed by different letters in the same column mean significant differences (*p* < 0.05).

## Data Availability

The data are available upon request.

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
