# Peer review of "Effects of Low-Molecular-Weight Glutenin Subunit Encoded by Glu-A3 on Gluten and Chinese Fresh Noodle Quality"

_foods, 2023, doi:10.3390/foods12163124_

Round 1

Reviewer 1 Report

In this manuscript, the authors reported effects of low-molecular-weight glutenin subunit encoded by Glu-A3 on gluten and also demonstrated their effects of Chinese fresh noodle quality. This work seems to be interesting and useful for this field. However, the following problems should be addressed:

Abstract: Present the most specific results. Insert numerical results related to the main findings of the work.

L33: How much gluten is in the wheat. Wheat total protein = gluten?

L35-36: In what processes?

L46-47: Why they are more important?

L67: "affect processing" - for what? This is a big generalization.

Table 1.:  Glu-A3ac?

L106-111: More details needed.

L187: Must be α < 0.05.

L238-239: What are the consequences. Explain please.

L258; L279; L288: Check the record “/”. Maybe a sign ( ) would be more appropriate?

L261-262: It is not clear.

L268-270: Where are the correlation coefficients?

L270-272: It is not clear.

Table 2: What is CV (%)?

L338-340: This is not a good comparison as different pastas have been tested. In addition, there are no sensory evaluation results in this paper.

The conclusion needs to be revised, in order to respond to the main objectives of the work. Conclusion: Needs to be more specific, it is too general.

L364-365: „provide useful information for wheat breeding“ – this information is not clear, expand.

References: Exclude some old references.

Grammer and spelling needs improvment.

Author Response

Response to Reviewer 1 Comments

Thank you very much for your positive and constructive comments on our paper (Number: foods-2508029). Those comments are all valuable and very helpful for revising and improving our paper. We have revised our manuscript according your comments and suggestion.

Below, I will detail how we revised the paper to address each of the comments by the reviewers, following the rank of the comments in the original decision letter. Changes in the revised manuscript are marked in highlighted red.

Point 1. Abstract: Present the most specific results. Insert numerical results related to the main findings of the work.

Response 1: Thank you for your suggestion. We retained the results with significant difference. These results comprehensively reflect the difference of gluten and Chinese fresh noodle quality among four NILs. The rank of gluten and Chinese fresh noodle quality was the result of comprehensive comparison of multiple parameters. Thus, we think that insert numerical results related to the main findings of the work was inappropriate. Please see abstract.

Point 2. Line 33: How much gluten is in the wheat. Wheat total protein = gluten?

Response 2: Thank you for your question. The relationship between wheat total protein and gluten have been added. Please see L32-34.

Point 3. L35-36: In what processes?

Response 3: Thank you for your question. We changed the processing quality to flour processing quality, and checked the full text. Please see L13, L37-38, L49, L69.

Point 4. L46-47: Why they are more important?

Response 4: Thank you for your question. We referred to the results of previous comparative studies. Previous research result showed Glu-A3 and Glu-B3 were more important compared with Glu-D3. Please see L47-49.

Point 5. L67: "affect processing" - for what? This is a big generalization.

Response 5: Thank you for your question. We changed the processing quality to flour processing quality. Please see L69.

Point 6. Table 1.: Glu-A3ac?

Response 6: Thank you for your question. There is no marker for direct identification of LMW-GS Glu-A3c. The marker Glu-A3ac can identified both LMW-GS Glu-A3a and Glu-A3c. Thus, two markers Glu-A3ac and Glu-A3a were used together for the identification of Glu-A3c.

Point 7. L106-111: More details needed.

Response 7: Thank you for your suggestion. More details have been added. Please see L110-118.

Point 8. L187: Must be α < 0.05.

Response 8: Thank you for your suggestion. In this study, when P<0.05, we decided there was a significant difference.

Point 9. L238-239: What are the consequences. Explain please.

Response 9: Thank you for your question. The consequences have been added. Please see L242-243.

Point 10. L258; L279; L288: Check the record “/”. Maybe a sign ( ) would be more appropriate?

Response 10: Thank you for your suggestion. When we use the sign () instead of the record /, some places will become inappropriate. Such as L267.

Point 11. L261-262: It is not clear.

Response 11: Thank you for your comment. We reorganized the language to make it clearer. Please see L264-266.

Point 12. L268-270: Where are the correlation coefficients?

Response 12: Thank you for your question. The significant difference of UPP% ranked as Glu-A3b>Glu-A3c>Glu-A3a/Glu-A3e. Similarly, the significant difference of GI also ranked as Glu-A3b>Glu-A3c>Glu-A3a/Glu-A3e. So, we think that the consistently significant difference between UPP% and GI indicated the well correlation. But we did not do a correlation analysis, the inference of correlation has been removed.

Point 13. L270-272: It is not clear.

Response 13: Thank you for your comment. We changed quality to content and index to make it clearer. Please see L272.

Point 14. Table 2: What is CV (%)?

Response 14: Coefficient of variation (CV) can compare the degree of dispersion between two sets of data. The greater the degree of dispersion, the stronger the influence on the result. Thus, LMW-GS mainly influence dough development time (DDT), stability time (ST), and farinograph quality number (FQN) in this study.

Point 15. L338-340: This is not a good comparison as different pastas have been tested. In addition, there are no sensory evaluation results in this paper.

Response 15: Thank you for your comment. Although different pastas have been tested, the Chinese noodle also was worth to test. Because Chinese noodle was softer, they have the different raw material needs. Meanwhile, near-isolated lines were the ideal material to study the difference among LMW-GS allelic variation, few of these special materials are used for such research. Sensory evaluation need to professional sensory evaluators. Because lacking professional sensory evaluators, there are no sensory evaluation results in this paper.

Point 16. The conclusion needs to be revised, in order to respond to the main objectives of the work. Conclusion: Needs to be more specific, it is too general.

Response 16: Thank you for your suggestion. We have revised the conclusions to make it more specific. Please see L362-374.

Point 17. L374-375: “provide useful information for wheat breeding” – this information is not clear, expand.

Response 17: Thank you for your suggestion. We have revised it. Please see L372-374.

Point 18. References: Exclude some old references.

Response 18: Thank you for your suggestion. Three old references (as follows) have been excluded.

references

Batey, I.; Gupta, R.; MacRitchie, F., Use of size-exclusion high-performance liquid chromatography in the study of wheat flour proteins: an improved chromatographic procedure. Cereal Chemistry 1991, 68, (2), 207-209. https://doi.org/10.1021/bp00008a015.

Irie, K.; Horigane, A. K.; Naito, S.; Motoi, H.; Yoshida, M., Moisture distribution and texture of various types of cooked spaghetti. Cereal chemistry 2004, 81, (3), 350-355. https://doi.org/10.1094/CCHEM.2004.81.3.350.

Edwards, N.; Gianibelli, M.; McCaig, T.; Clarke, J.; Ames, N.; Larroque, O.; Dexter, J., Relationships between dough strength, polymeric protein quantity and composition for diverse durum wheat genotypes. Journal of Cereal Science 2007, 45, (2), 140-149. https://doi.org/10.1016/j.jcs.2006.07.012.

Point 19. Comments on the Quality of English Language

Grammer and spelling need improvement.

Response 19: Thank you for your suggestion. We checked for grammar and spelling errors using the Spelling and Grammar function in Word.

Once again, thank you very much for the time and efforts you expanded on our manuscript.

Reviewer 2 Report

The article entitled "Effects of low-molecular-weight glutenin subunit encoded by Glu-A3 on gluten and Chinese fresh noodle quality" is well-structured, with relevant analysis and techniques employed. The work provides useful information for wheat breeding and processing of high-quality Chinese fresh noodle. The Introduction section provides background about the topic. The experimental design is adequately discussed. Interesting results were obtained by suitable methods and data interpretation is reliable, but the discussions must be improved with the latest references supporting the results of the manuscript. The Conclusion section was supported by the results, and provides a good conclusion for the study.

 Manuscript can be further improved taking following points into consideration.

 Line 104:  …. markers instead of … molecular makers 

The material and methods section presented a high diversity of methods, but some of these need a briefly describe, e.g., 2.2.2, 2.2.3, and 2.3 methods at lines 106, 109, and 112 respectively.

Line 191: molecular markers instead of molecular makers

Line 255-256: … no significant difference (P > 0.05) was found among the four NILs (Figure 4A and 4B) instead of …. no significant difference were found among the four NILs (P<0.05, Figure 4A and 4B)

Table 4. Please verify! (g×s) instead of (g.s), (g.sec)

Line 289: Meanwhile, the relationship …. instead of Meanwhile, The relationship ….

Author Response

Response to Reviewer 2 Comments

Thank you very much for your positive and constructive comments on our paper (Number: foods-2508029). Those comments are all valuable and very helpful for revising and improving our paper. We have revised our manuscript according your comments and suggestion.

Below, I will detail how we revised the paper to address each of the comments by the reviewers, following the rank of the comments in the original decision letter. Changes in the revised manuscript are marked in highlighted red.

Point 1. Line 104:  …. markers instead of … molecular makers

Response 1: Thank you for your suggestion. We have used markers instead of molecular makers, and check the full text. Please see L103, L106, L199, L202 et al.

Point 2. The material and methods section presented a high diversity of methods, but some of these need a briefly describe, e.g., 2.2.2, 2.2.3, and 2.3 methods at lines 106, 109, and 112 respectively.

Response 2: Thank you for your suggestion. More details have been added. Please see L110-124.

Point 3. Line 191: markers instead of molecular makers

Response 3: Thank you for your suggestion. We used markers instead of molecular makers, and check the full text.

Point 4. Line 255-256: … no significant difference (P > 0.05) was found among the four NILs (Figure 4A and 4B) instead of …. no significant difference were found among the four NILs (P<0.05, Figure 4A and 4B)

Response 4: Thank you for your suggestion. We have made changes and reviewed the full text. Please see L237 and L264.

Point 5. Table 4. Please verify! (g×s) instead of (g.s), (g.sec)

Response 5: Thank you for your suggestion. We used the unit of chewiness incorrectly, and now it has been modified. Please see Table 4.

Point 6. Line 289: Meanwhile, the relationship …. instead of Meanwhile, The relationship ….

Response 6: Thank you for your suggestion. We have used Meanwhile, the relationship …. instead of Meanwhile, The relationship …., and check the full text. Please see Line 296.

Once again, thank you very much for the time and efforts you expanded on our manuscript.

Round 2

Reviewer 1 Report

-